# The Potential Selective Cytotoxicity of Poly (L- Lactic Acid)-Based Scaffolds Functionalized with Nanohydroxyapatite and Europium (III) Ions toward Osteosarcoma Cells

**DOI:** 10.3390/ma12223779

**Published:** 2019-11-18

**Authors:** Mateusz Sikora, Klaudia Marcinkowska, Krzysztof Marycz, Rafał Jakub Wiglusz, Agnieszka Śmieszek

**Affiliations:** 1The Department of Experimental Biology, The Faculty of Biology and Animal Science, University of Environmental and Life Sciences Wroclaw, 38 C Chelmonskiego St, 50–630 Wroclaw, Poland; mateusz.sikora@upwr.edu.pl (M.S.); klaudia.marcinkowska@upwr.edu.pl (K.M.); krzysztof.marycz@upwr.edu.pl (K.M.); 2International Institute of Translational Medicine, Jesionowa 11 St, 55–124 Malin, Poland; 3Collegium Medicum, Institute of Medical Science, Cardinal Stefan Wyszyński University (UKSW), Wóycickiego 1/3, 01-938 Warsaw, Poland; 4Institute of Low Temperature and Structure Research, Polish Academy of Sciences, Okolna 2, 50–422 Wroclaw, Poland; r.wiglusz@intibs.pl; 5Centre for Advanced Materials and Smart Structures, Polish Academy of Sciences, Okolna 2, 50–950 Wroclaw, Poland

**Keywords:** biomaterials, polylactide, nanohydroxyapatite, europium (III) ions, osteosarcoma, theranostics, regenerative medicine, bio-imaging

## Abstract

Osteosarcoma (OSA) is malignant bone tumor, occurring in children and adults, characterized by poor prognosis. Despite advances in chemotherapy and surgical techniques, the survival of osteosarcoma patients is not improving significantly. Currently, great efforts are taken to identify novel selective strategies, distinguishing between cancer and normal cells. This includes development of biomimetic scaffolds with anticancer properties that can simultaneously support and modulate proper regeneration of bone tissue. In this study cytotoxicity of scaffolds composed from poly (L-lactic acid) functionalized with nanohydroxyapatite (nHAp) and doped with europium (III) ions—10 wt % 3 mol % Eu^3+^: nHAp@PLLA was tested using human osteosarcoma cells: U-2 OS, Saos-2 and MG-63. Human adipose tissue-derived stromal cells (HuASCs) were used as non-transformed cells to determine the selective cytotoxicity of the carrier. Analysis included evaluation of cells morphology (confocal/scanning electron microscopy (SEM)), metabolic activity and apoptosis profile in cultures on the scaffolds. Results obtained indicated on high cytotoxicity of scaffolds toward all OSA cell lines, associated with a decrease of cells’ viability, deterioration of metabolic activity and activation of apoptotic factors determined at mRNA and miRNA levels. Simultaneously, the biomaterials did not affect HuASCs’ viability and proliferation rate. Obtained scaffolds showed a bioimaging function, due to functionalization with luminescent europium ions, and thus may find application in theranostics treatment of OSA.

## 1. Introduction

Osteosarcoma (OSA) is well-examined neoplasm, which belongs to the group of bone-tissue sarcomas producing osteoid. It is predominantly diagnosed in animals, but also frequently develops in humans. Tumor lesions consistently locate in the long bones e.g., femur, tibia or humerus, but sometimes they are revealed in unusual places e.g., ribs or pelvis bones [1,2]. Regarding morbidity, scientists distinguish two essential incidence points of OSA—during the juvenile period and in adults older than 65 years of age. The main reasons of the disease progression are: (i) bone growth spurt in puberty or (ii) natural changes in bone structure related to the aging of individual [3,4]. 

The data from the last five years indicate an average incidence of OSA of 3.4 cases per million people a year. Furthermore, the chance of 10-year survival is estimated at 66% in the case of non-metastatic cancer, and only 24% in the case of scattered metastases [5,6]. Due to high malignancy of this tumor type, the treatment is still a huge challenge to modern medicine. However, the possibility of patients’ survival could grow significantly through the use of new, alternative methods of treatment, including those based on biomaterials application [7,8]. Nowadays, surgical treatment combined with chemotherapy is considered as the most adequate procedure in osteosarcoma treatment. The surgical resection of a tumor can be radical, with amputation of a whole limb. However, limb-sparing surgery (LSS) is also more frequently performed, mainly due to development in surgical procedures, and progress in adjuvant chemotherapy [9]. Additionally, no survival advantage to amputation over limb-salvage procedures was found in a large retrospective studies presented by Simon et al. (1986) and Gherlinzoni et al. (1992) [10,11]. Given this fact, the limb salvage surgery is more frequently suggested, especially when proper surgical margins can be achieved [12,13,14]. Primary chemotherapeutics used for the treatment of osteosarcoma are: adriamycin (ADM), cisplatin (DDP), high-dose methotrexate (HD-MTX), ifosfamide (IFO) and epirubicin (EPI) [15]. These drugs improve the survival rate of patients with osteosarcoma, however their efficacy is usually associated with the application of high-intensity doses. Such a treatment strategy increases the incidence of chemotherapy complications and risk of fatal toxicity [16]. The development of chemoresistance is also the factor that limits the effectiveness of drugs used for OSA treatment. For example, HD-MTX, the application of which is associated with great variation in individual responses, may induce primary or secondary drug resistance [17]. Further, chemotherapy regimens used for OSA treatment are unspecific, targeting tumor cells but also normal, healthy tissue. In particular, it was shown that chemotherapy used for OSA management may affect the hair follicle stem cells [18], but also can cause bone marrow suppression or affect cells lining the digestive tract [16,19]. 

Despite the fact that various chemotherapeutic regimens are recommended, we are still seeking new options of treatment as the ideal drug for osteosarcoma does not exist [2,20]. Combination of modern and innovative methods basing on selective therapies and tissue engineering methods are strongly considered in terms of osteosarcoma treatment [21,22]. The therapeutic strategies are aimed at developing multifunctional scaffolds - not only supporting functional bone regeneration but also delivering anti-cancer agents. The polymer scaffolds are considered as the most sufficient delivery systems for various bioactive agents [23,24]. Poly(L-lactic acid) (PLLA) draws attention in terms of biomedical field usage, due to its high biocompatibility, controllable biodegradability, thermal plasticity and suitable mechanical properties. Polylactides are approved by the US Food and Drug Administration (FDA), which emphasizes their potential for clinical application in various medical condition including orthopedic [25,26].

In the present study, using model of human osteosarcoma cell lines, selective function of biodegradable poly (L-lactic acid) (PLLA) scaffolds functionalized with nanohydroxyapatite (nHAp 10 wt %) doped with europium (III) ions (3 mol% Eu^3+^ ions) was tested. The cytocompatibility of the PLLA/nHAp matrices was characterized previously [27] using model of human adipose-derived multipotent stromal cells (HuASC). It has been previously shown that obtained scaffolds improves viability of HuASC which correlates with increased metabolic activity of cells. Furthermore, we have functionalized PLLA/nHAp matrices with europium (10 wt % 3 mol % Eu^3+^: nHAp/PLLA), and obtained scaffolds showing pro-osteogenic properties (data ahead of publishing). Additionally, the presence of luminescent europium allowed us to obtain scaffolds with potential application in theranostics [28]. During preliminary studies, cytocompatibility of 10 wt % 3 mol % Eu^3+^: nHAp/PLLA matrices was also tested using a well established model of osteosarcoma cell lines, including U-2 OS, Saos-2 and MG-63. Those cell lines are being generally used in terms of cytocompatibility determination of various biomaterials due to the fact that poses features of osteoblast-like cells and are capable to undergo osteoblastic differentiation [29,30,31,32,33,34]. The screening assays (data not shown) indicated that 10 wt % 3 mol % Eu^3+^: nHAp/PLLA has an adverse effect toward OSA cell lines and inhibits their growth determined by MTS assay. Thus in this study, profound characterization of potential selective activity of scaffolds was performed. 

Nanohydroxyapatite was previously described as an agent with anti-cancer properties, with cytotoxic effect toward osteosarcoma cells [35]. For instance, it was shown that nanohydroxyapatite induce apoptosis and decrease proliferation of murine osteosarcoma cell line K7M2 and human osteosarcoma cells (MG-63) [36,37]. The cytotoxic effect of nanohydroxyapatite was also confirmed in a model of human liver tumor cells (line HepG2) and leukemia cell line P388 [38,39,40]. Research confirmed that the programmed cell death caused by nHAp is related to the intrinsic pathway, e.g., dependent on mitochondria and the release of caspases. Nanohydroxyapatite changes the potential of the mitochondrial membrane, which affects the initiation of reactions leading to annihilation of the cell [41]. Europium ions can also be important element of engineered carrier during regeneration of bone tissue in the process of osteosarcoma treatment [28]. Moreover, functionalization of scaffolds with europium ions provides the opportunity to observe the carrier dynamics simultaneously monitoring cells fate and process of tissue healing (theranostics). Furthermore, it was shown that europium ions improve the bone mineralization and doxorubicin release [19], which make it a valuable component of designed biomaterial and gives opportunity to use the scaffold as a drug delivery system in oncology therapies. 

Bearing in mind high cytocompatibility and pro-osteogenic properties of 10 wt % 3 mol % Eu^3+^: nHAp/PLLA matrices determined using the model of HuASC, and simultaneous cytotoxicity of this scaffolds toward human osteosarcoma cell lines, we decided to elucidate their potential selective activity. The analysis included determination of scaffolds influence on cell viability (apoptosis profile) and caspase activation. We evaluated expression of genes associated with cells’ survival and proliferation such as *P53, BAX, BCL-2, C-MYC* or *MMP-7*, and established microRNA levels associated with tumor cells’ viability and osteogenic activity. The study included also profound characteristic of cells morphology and growth pattern in cultures with the scaffolds. 

To the best of our knowledge this study describes for the first time therapeutic potential of Eu^3+^: nHAp/PLLA scaffolds in terms of anticancer applications. Promising selectivity of the scaffolds may find clinical translation in OSA treatment exerting anti-proliferative and pro-apoptotic effects on osteosarcoma cells and simultaneously improving viability and metabolic potential of progenitor cells.

## 2. Materials and Methods 

### 2.1. Biomaterial Preparation

The nanocrystalline powders of hydroxyapatite and hydroxyapatite doped with Eu^3+^ ions were prepared by the microwave-stimulated hydrothermal method. Concentration of the europium (III) ions was set to 3 mol % in proportion to the overall molar content of Ca^2+^ ions. As starting substrates have been used analytical grade of Ca(NO_3_)_2_∙4H_2_O (99+% Acros Organics, Argenta, Poznan, Poland), NH_4_H_2_PO_4_ (99.995% Alfa Aesar, Argenta, Poznan, Poland), Eu_2_O_3_ (99.99% Alfa Aesar, Argenta, Poznan, Poland) and NH_3_∙H_2_O (99% Avantor, Gliwice, Poland) for pH adjustment. In the case of hydroxyapatite doped with Eu^3+^ ions, the stoichiometric amounts of Eu_2_O_3_ were digested in an excess of the HNO_3_ (ultrapure, Avantor, Gliwice, Poland) to obtain water-soluble nitrates and then europium(III) nitrates were re-crystalized three times. Subsequently, calcium nitrate was dissolved in deionized water together with europium nitrate. The suitable amount of ammonium phosphate dibasic was added to the mixture leading to fast precipitation of the by-product. The pH of the dispersion was modulated to 10 by adding ammonia, transferred into the Teflon vessel and set in the microwave reactor (ERTEC MV 02–02, Ertec, Wroclaw, Poland). After 90 min of the microwave stimulated hydrothermal processing at 280 °C and under autogenous pressure of 60 atm., a nanocrystalline powder was obtained. Powder was washed with de-ionized water several times and dried at 70 °C for 24 h in order to get the final product. In order to receive a well-crystallized product and remove residual amorphous phase thermal treatment in the temperature at 500 °C was applied. In the next step the nHAp/PLLA biocomposite was fabricated in the procedure described by Wiglusz and co-workers. Moreover, the physicochemical characteristics of examined biomaterial (10 wt % 3 mol % Eu^3+^: nHAp/PLLA were also characterized previously [27].

### 2.2. Cell Cultures

Three human osteosarcoma cell lines U-2 OS, Saos-2 and MG-63 were used for the experiment. The cell lines derived from the European Collection of Authenticated Cell Cultures (ECACC, Sigma-Aldrich, Poznan, Poland). Additionally, in order to analyze selective cytotoxicity of biomaterial the adipose-derived human mesenchymal stromal cells (HuASC) were used. All cell lines were propagated in sterile conditions using CO_2_ incubator with constant conditions: 37 °C, 5% CO_2_ and 95% humidity. For the test cells were inoculated in 24-well dishes covered with biomaterial at density equal 30,000 cells per well, suspended in a 0.5 mL of complete growth medium (CGM). CGMs consisted of McCoy’s 5A for U-2 OS and Saos-2, Eagles’ Minimum Essential Medium for MG-63 or Ham’s F-12 nutrient mixture for HuASC. All media were supplemented with 10% of fetal bovine serum (FBS) and 1% of antibiotic solution containing 10,000 units penicillin and 10 mg streptomycin/mL (BioReagent). Upon reaching approximately 90% confluence the cultures were washed in Hank’s Balanced Salt Solution (HBSS) and trypsinized for 10 min at 37 °C (StableCell™ Trypsin Solution). After that, the cells were centrifuged (300× *g*, 4 min) and suspended in fresh medium. Reagents used for cell culture i.e., basal media, FBS, antibiotics, salts and trypsin derived from Sigma-Aldrich (Poznan, Poland). 

### 2.3. The Experiment

Cells were inoculated in 24-well dishes at density equal 30,000 cells per well, suspended in a 0.5 mL of CGM. In experimental cultures cells were seeded on wells covered with biomaterial. Cells propagated on polystyrene served as a control of the experiment. Cultures were propagated for 72 h. After the experiment cultures were processed for further analysis. The experiment was repeated three times.

### 2.4. Analysis of Biomaterial Impact on Cells’ Morphology and Eu^3+^ Ions-Doped nHAp Colocalization

The cells morphology was evaluated using confocal microscope. The analysis was performed after 72 h of cells propagation onto biomaterial surface. Before microscopic observations the CGM was removed, cultures were washed with HBSS and then fixed with 4% paraformaldehyde (PFA) solution for 10 min at room temperature. Further, cells were permeabilized with 0.2% phosphatase buffered saline (PBST) with Tween 20 and actin cytoskeleton was stained with phalloidin solution (Atto 488, Sigma-Aldrich, Poznan, Poland). The dye was prepared at the dilution of 1:800 in HBSS for 40 min at 37 °C. Subsequently, the preparations were closed on microscope slides using Mounting Medium with DAPI (4′,6-diamidino-2-phenylindole; Thermo Fisher Scientific, Warszawa, Poland). Samples were observed under confocal microscope (Leica TCS SPE, Leica Microsystems, Wetzlar, Germany). Microphotographs were captured at 630 × magnification.

### 2.5. Analysis of Biomaterial Impact on Cells’ Adhesion and Intercellular Interactions

The cells’ adhesion, cell–cell and cell–biomaterial interactions were evaluated using scanning electron microscope (SEM). After experiment, cultures were washed with HBSS, and fixed with 4% PFA solution, for 10 min. After that samples were dehydrated in a graded ethanol series (concentrations from 50% to 100%, changed every 5 min). Then, specimens were spurred with gold particles for 500 s (Edwards Pirani 501 Scancoat Six, Edwards Laboratories, Milpitas, CA, United States). The SEM Evo LS 15 Zeiss (Carl Zeiss Microscopy, Jena, Germany) was used for capturing the images. The imaging was carried out using the SE1 detector at 10 kV and 4000 × magnification.

### 2.6. Analysis of Biomaterial Impact on Cells’ Metabolic Activity

#### 2.6.1. MTS Test

MTS assay was used to determine the metabolic activity of cells after 72 h of culture onto biomaterial surface. For this purpose, cells were harvested using trypsin solution (as indicated in Section 2.2.) and inoculated on 96-well dishes in triplicates. Then, 20 µL of MTS reagent was added to the cells and cultures were incubated for 2 h at 37 °C. The assessment of metabolic activity was analyzed by measuring the absorbance at 490 nm. The complete CGM with the addition of MTS reagent was used as negative sample (blank).

#### 2.6.2. Mitochondria Depolarization Status

Analysis of the mitochondrial membrane activity was performed using the Muse^TM^ Mitopotential Assay Kit (Merck^®^; cat. no.: MCH100110, Poznan, Poland). The staining procedure was carried out in accordance with the manufacturer’s instructions. After the experiment cells were trypsinized (as indicated in Section 2.2.). After centrifugation obtained pellet of cells was resuspended in 100 μL of buffer supplied with the kit and stained with the Muse™ MitoPotential dye. The staining solution was prepared in accordance with the manufacturer’s protocol at the dilution of 1:1000 in the analysis buffer. The dye solution was added to the cells suspension at volume equal 95 μL and incubated for 20 min in CO_2_ incubator (37 °C, 95% humidity, 5% CO_2_). Subsequently, cells were stained by adding 5 μL of 7-AAD dye. The cells were incubated for 5 min at room temperature. Mitochondrial membrane activity was analyzed using a Muse^TM^ Cell Analyzer. Each analysis was performed in three replicates.

### 2.7. Analysis of Biomaterial Impact on Caspases Activation

Analysis of caspase activation was performed using the Muse™ MultiCaspase Kit (Merck^®^; cat. no.: MCH100109, Poznan, Poland). The assay was carried out in accordance with the protocol published by the manufacturer. Prior to the assay, cells were harvested as described above (Section 2.2.). Caspase buffer supplied with the kit was diluted 10 times using diethylpyrocarbonate-treated water (DEPC-treated water). MultiCaspase Reagent Stock Solution was prepared by adding 50 μL DMSO to the included MultiCaspase Reagent dye. MultiCaspase Reagent Working Solution was prepared at the dilution of 1:160 in 1 × PBS. Caspase 7-AAD Working Solution was prepared by adding 2 μL of the Muse™ Caspase 7-AAD dye to 148 μL of 1 × Caspase Buffer. After trypsynization, 50 μL of cells were suspended in 5 μL of the Muse^TM^ MultiCaspase Working Solution reagent. After 30 min of incubation in a CO_2_ incubator (37 °C, 95% humidity, 5% CO_2_) 150 μL Muse^TM^ 7-AAD Working Solution was added. The activity profile of the caspases was determined using the Muse^TM^ Cell Analyzer. Each analysis was performed in triplicate.

### 2.8. Analysis of Biomaterial Impact on Genes Expression Involved in Apoptosis and Cell Cycle 

The transcript levels of selected genes were analyzed using quantitative reverse transcriptase real-time polymerase chain reaction (RT-qPCR). After the experiment, cells were harvested using trypsin solution as indicated above (Section 2.2.). Further, cells were homogenized using 1 mL of TRI Reagent^®^. The RNA isolation was carried out in accordance with manufacturer’s instruction, which is method described by Chomczyński and Sacchi [42]. Precision^TM^DNase kit (Primerdesign Ltd., Blirt, Gdansk, Poland) was used to digest the genomic DNA residues. The cDNA was obtained from 100 ng of RNA using Tetro cDNA Synthesis Kit (Bioline Reagents Ltd., London, United Kingdom). For the reaction1 μL of cDNA was used, analysis was performed in a total volume of 10 μL using SensiFast SYBR^®^ and Fluorescein Kit (Bioline Reagents Ltd., London, United Kingdom). All primers were used in concentration equal 400 nM (Table 1.). The qPCR was performed using CFX Connect^TM^ Real-Time PCR Detection System (Bio-Rad Polska Sp. z.o.o., Warsaw, Poland). The maximum relative quantification (RQ_max_) of gene expression was calculated in relation to a housekeeping gene (GAPDH/glyceraldehyde 3-phosphate dehydrogenase) using 2^−∆∆Ct^ algorithm. 

### 2.9. Analysis of Biomaterial Impact at the miRNA Level

The levels of selected miRNA were analyzed using RT-qPCR. After experiments cultures were harvested as described above and cells were homogenized using 1 mL of TRI Reagent^®^. Precision^TM^DNase kit (Primerdesign Ltd., Blirt, Gdansk, Poland) was used to digest the genomic DNA residues. The cDNA was obtained from 250 ng of RNA using Mir-X™ miRNA First-Strand Synthesis Kit (Takara Clontech Laboratories, Biokom, Poznan, Polska). Before the qPCR, samples were diluted 10-fold using DEPC water. The reaction was performed as described above (Section 2.8) and using protocol described previously [43]. The maximum relative quantification (RQ_max_) of gene expression was calculated in relation to a snU6 gene using 2^−∆∆Ct^ algorithm. The primers used for the reaction are listed in Table 2. The sequence of primers for the snU6 are not included, due to the fact that the primers are supplied with the kit. 

### 2.10. Statistical Analysis

Statistical analysis was performed using GraphPad Prism 8 (GraphPad Software, San Diego, CA, USA) Comparisons between the experimental groups and control groups were carried out using the Student’s *t*-test. Differences were considered statistically significant at p < 0.05, using the designations: * *p* < 0.05, ** *p* < 0.01, *** *p* < 0.001. Obtained results are presented on a statistical graphs as mean values obtained in three independent repetitions, while whiskers represent standard deviation (± SD) obtained for the assays. 

## 3. Results

### 3.1. Biomaterial Impact on Cells’ Morphology 

Observations performed using confocal microscope revealed that osteosarcoma cell lines cultured in the presence of biomaterial had poorly developed cytoskeleton and did not form an integral monolayer, which was characteristic for cultures on polystyrene surface. The alteration of actin cytoskeletal organization was associated with weakened intercellular interactions (cell–cell contact). The number of cells attached to the biomaterial was lowered, what can be observed based on nuclei distribution. Similarly, the number of progenitor cells (HuASC) was also reduced in cultures propagated on the biomaterial, however, unlike for osteosarcoma cells, no significant changes were noticed in terms of actin organization. In HuASCs intercellular spaces were less visible than in osteosarcoma cell cultures, which indicates the presence of cell-cell and cell-biomaterial interactions. HuASCs showed typical fibroblast–like morphology (Figure 1).

### 3.2. Biomaterial Impact on Cells’ Adhesion and Intercellular Interaction

The analysis revealed that all cells used in the experiment interact with the biomaterial. Besides cell-biomaterial contact, the presence of cell-cell interactions was also evident. The scanning electron microscopy (SEM) analysis confirmed biomimetic character of the scaffold (Figure 2). 

### 3.3. Analysis of Cells Viability Based on Caspase Activation

The analysis revealed that biomaterials induce the activation of caspase in all tested osteosarcoma cell lines. The comparative analysis between control and experimental cultures showed significant increase of caspase-positive cells in osteosarcomas propagated in the presence of biomaterial. The carrier has no significant impact on HuASC caspase activation (Figure 3).

### 3.4. Biomaterial Impact on Cells’ Metabolic Activity

The metabolic activity was determined based on the results of MTS assays. The analysis revealed that osteosarcoma cells U-2 OS and Saos-2 had decreased metabolic activity measured with MTS. The analysis of mitochondrial membrane potential confirmed this result showing that U-2 OS, Saos-2 cultured on biomaterials, were characterized by lowered metabolism reflected by mitochondrial membrane depolarization. In turn, the MG-63 propagated with the scaffold had improved mitochondrial membrane potential what was determined using both MTS and cytometric measurement. The results of MTS showed that metabolic activity of HuASC is not affected by the presence of the scaffold, however, the analysis of mitochondrial membrane potential indicated an increased population of cells with depolarized mitochondrial membrane (Figure 4).

### 3.5. Biomaterial Impact on Transcriptional Activity

The cytotoxicity of PLLA/ Eu^3+^: nHAp biomaterial was also evaluated based on mRNA profile of genes involved in apoptosis and cell cycle (Figure 5 and Figure 6). Obtained results indicated that mRNA level for anti-apoptotic gene *P21* was decreased in Saos-2 and MG-63. Moreover, the expression of *P53* was increased in Saos-2 and HuASC, but decreased in MG‑63 line. The transcript level of *APAF* was significantly decreased in Saos-2 and MG‑63. Additionally, we observed that key pro-apoptotic gene *BAX* was increased in MG-63, but decreased in U-2 OS and Saos-2. In turn, the mRNA level of anti-apoptotic *BCL-2* was decreased in U-2 OS and MG-63, but increased in Saos-2. The mRNA levels of *Cyclin D*, *C‑MYC* and *Survivin*, which are determinants of proliferation, were significantly increased in the HuASC line. However, in U-2 OS and Saos-2 cell lines the levels of *Cyclin D* transcripts were decreased, while in MG-63 it was increased. The *C-MYC* was decreased in U-2 OS and MG‑63, but increased in Saos-2. The *Survivin* was significantly decreased in Saos-2 and MG‑63, but increased in U-2 OS.

The mRNA levels of metalloproteinase 7 (*MMP-7*) and metalloproteinase 14 (*MMP-14*) were seen to be lower in MG-63 as a result of their propagation on biomaterial. However, the *MMP-7* was increased in Saos-2 and HuASC. The *MMP-14* was also increased in U-2 OS. The biomaterial caused an effect on the expression of caspases. The transcript level of *Casp3* was increased in Saos-2 and HuASC, but decreased in U-2 OS and MG-63. The mRNA level of *Casp6* was only significantly increased in MG-63. Except the HuASCs, all tumor cell lines had increased level of *Casp8*. Although, the mRNA level of *Casp9* was increased in Saos-2, it was also decreased in U-2 OS and HuASCs.

### 3.6. Biomaterial Impact at the miRNA Level

Examined biomaterial showed slight impact on the miRNA levels. During propagation onto biomaterial surface, only in MG-63 tumor cell line the levels of *miR-21a-5p, miR-124-3p, miR-203a-3p, miR-320-3p* were significantly decreased. No significant changes in miR profile were observed in the other cell lines (Figure 7).

## 4. Discussion

Modern regenerative medicine is undeniably one of the fastest developing scientific fields, which gives a great hope to the patients that suffer from serious diseases or injuries. Current efforts are put into developing novel composites tailored to meet the great needs of regenerative medicine and oncology [44,45]. The opportunity to design alternative treatment methods gives a real chance to restore damaged tissues and prolong patients’ healthy life. The use of biomimetic multifunctional biomaterials gives a chance to save a limb and improve patient regeneration potential after tumor resection. Innovative biomaterials, especially polymer-based are widely considered as therapeutic strategy for bone tumor treatment, including osteosarcoma [46,47,48]. This malignant tumor is still characterized by high mortality and drug resistance [6]. Anti-cancer drugs commonly used for osteosarcoma treatment, such as doxorubicin, cisplatin brings many negative side effects [2,49], while radiotherapy is considered as palliative treatment method, due to radiation-induced apoptosis resistance [49,50]. The most recent studies are aimed at designing biodegradable scaffolds that could be used for cancer regenerative medicine. The scaffolds are tailored to obtain proper recovery of tissue and restore its function, but also to inhibit tumor cell proliferation and tumor occurrence [51]. The development of personalized therapies and targeted (selective) strategies is essential in terms of osteosarcoma treatment. Most recently, Raucci et al., developed exfoliated black phosphorous (2D bP) substrates characterized by selective properties. The 2D bP material showed anti-proliferative activity toward Saos-2 cell line, and induced their apoptosis by increasing the production of reactive oxygen species (ROS). In turn 2D bP promoted proliferation and differentiation of human pre-osteoblasts (HOb) exerting protective effect against oxidative stress species [52]. Polymer-based scaffolds gain special attention in terms of cancer therapies also as a drug-delivery systems that can improve drug release and pharmacokinetics. For example, Murugan et al. developed nano-hydroxyapatite reinforced with a xylitol based poly(xylitol sebacate) PXS co-polymer loaded with capsaicin. The scaffold mechanism of action was associated with increased production of reactive oxygen species that induced apoptosis of Saos-2 cells [53]. Moreover, microporous poly-ε-caprolactone (PCL) scaffolds were obtained by Palama et al. to sustain the local release of anti-inflammatory drug dexamethasone (DXM) [48]. Recently composites, based on biopolymers and calcium phosphates with inclusion of magnetic nanoparticles loaded with doxorubicin, were described as strategy suitable for combination therapy (radiotherapy followed by chemotherapy) of malignant bone tumors [54]. Given the above examples, polymer-based carriers are considered as a tool that can be used to provide selective targeting of tumor sites without significant damage to the viability of healthy tissues. 

In the current study we have examined the cytotoxicity of scaffold consisted of poly (L‑lactic acid) (PLLA) functionalized with nanohydroxyapatite (nHAp) doped with europium (III) ions (Eu^3+^) – 10 wt % 3 mol % Eu^3+^: nHAp/PLLA. PLLA is defined as a fully biodegradable and biocompatible polymer, widely used in terms of biomedical applications [27,55]. nHAp as a main active component of engineered scaffold, was characterized as cytotoxic towards different tumor cells, including osteosarcoma [56,57]. However, the previous examination of nHAp particles showed also their osteoimmunomodulatory and pro-osteogenic properties toward human adipose-delivered stromal cells (HuASC) [27,28]. It could be a result of selective and dual-targeted properties of nanohydroxyapatite, which is desirable in regenerative medicine [35]. Thus, nHAp as a component of designed biomaterial, can provide effective selectivity of scaffolds, manifested in toxicity towards tumor cells and beneficial properties towards HuASCs. The addition of Eu^3+^ gives a possibility to use biomaterial in theranostics and bio-imaging. In addition, it was previously shown, that europium ions have positive effects towards HuASCs (adipose-derived stem cells) and BMSCs (bone marrow stem cells) which correlated with improved viability, cell growth and osteogenic potential [28,57]

In this study, cytotoxicity of scaffolds was determined using three human osteosarcoma cell lines, including U-2 OS, Saos-2 and MG-63. Additionally, human adipose-derives stromal cells were used as a reference model to this experiment. We have shown that growth pattern of all osteosarcoma cell lines is deteriorated in cultures on the scaffold. The result is in good agreement with Vohra et al. (2008) study, who showed loss of cell integrity in cultures of MG-63 and Saos-2 propagated on collagen scaffolds coated with hydroxyapatite [34]. In turn, we have not observed changes in growth pattern of HuASC maintained on the scaffold. Cells not only retained proper fibroblast morphology, but also improved cell–cell contact. The nHAp was previously identify as a component that improves cytocompatibility of PLLA scaffolds [27], but also various other scaffolds including gellan gum (GG) spongy-like hydrogels [58]. Using the model of HuASC, Bastos et al. (2019) indicated that nHAp may be a key factor promoting progenitor cells’ contact and thus promoting cells viability. We have previously shown, that europium ions may also affect the survival potential of progenitor cells [28]. In this study we have shown that europium is internalized by the cells locating in perinuclear region, what is in line with our previous observations [28] and other studies [38].

Our results indicated that obtained scaffolds modulate metabolic activity of both tumor and stromal cells. Measurement of mitochondrial metabolic rate, as well as analysis of mitochondrial membrane potential revealed that the scaffold significantly decreases metabolism of U-2 OS and Saos-2. Both nHAp and europium might be responsible for the cytotoxic effect observed toward those two osteosarcoma cell lines. The results obtained are in good agreement with studies presented by Cai et al. (2007) who showed that hydroxyapatite crystals affects the metabolic activity of U-2 OS [56]. Additionally, Seyedmajidi et al. [59] showed that hydroxyapatite/bioactive glass (HA/BG) at certain concentration may have a cytotoxic effect, and decrease metabolic potential of Saos-2. In turn, in this study we have shown that mitochondrial metabolism of MG-63 is accelerated in cultures on 10 wt % 3 mol % Eu^3+^: nHAp/PLLA scaffold. We did not observe a cytotoxic effect of the scaffold toward MG-63, which stays in line with studies Ito et al. [60]. In this research authors indicated that hydroxyapatite (HAp) derived from PEG-based composites is a factor that enhance metabolic activity of MG-63, what resulted in increased alkaline phosphatase (ALP) activity. Further, Begam et al. [61] showed that both pure nHAp as well as zinc doped nHAp promotes the adhesion and metabolic activity of MG-63. 

We have also shown that 10 wt % 3 mol % Eu^3+^: nHAp/PLLA inflects the metabolic activity of HuASC. The analysis of MTS assay indicated that scaffolds slightly improves the metabolic activity of HuASC, what confirmed our previous data [28], however this study revealed that mitochondrial membrane potential of HuASC decreases in cultures on 10 wt % 3 mol % Eu^3+^: nHAp/PLLA. This might be associated with pro-osteogenic properties of the scaffold that we revealed most recently (data ahead of publishing). In particular, Pietilä et al. showed that during osteogenic differentiation of bone marrow-derived mesenchymal cells the mitochondrial membrane potential decreases, while abundance of rough endoplasmic reticulum is reduced, however without influence on the cells’ viability [62]. This is in good agreement with the results regarding caspases profile. Our study indicated that viability of the progenitor cells is not affected by the scaffold presence—caspase activity was comparable in control and experimental conditions. In turn, in all tested osteosarcomas propagated on the scaffold, the increased activity of caspases was observed, indicating on selective cytotoxicity of obtained matrices. This might be associated with the anti-cancer properties of nHA, but also tumor-selective properties of lanthanides. For instance, LaCit and CeCl3 were shown to inhibit the proliferation of cancer cells but not human embryonal fibroblasts [63,64].

The analysis of transcript levels determined for genes associated with apoptosis revealed that the apoptosis process noted in osteosarcoma cell lines may depend on various factors, however in all osteosarcoma cell lines we noted increased mRNA level for caspase 8 (*Casp8*) which confirms studies published by Seki et al. who revealed that *Casp8* is a key molecule in the earliest stage of apoptosis induced by cisplatin in human osteosarcoma cells (HOS). Additionally, we noted an increased *Bax/Bcl-2* ratio in all osteosarcomas cell lines, however significant results were noted only for Saos-2 and MG-63 cell line. Results obtained indicate on the possibility for cross-talk between extrinsic and intrinsic apoptosis signaling pathways what may be essential in amplifying the apoptosis cascade [65]. The Saos-2 and MG-63 cell lines were also characterized by decreased mRNA levels for survivin. The same profile was also noted in osteosarcoma tissues after cisplatin treatment [65]. 

In turn, the analysis of the miRNA profile showed that obtained scaffolds do not alter the levels of *miR-21a-5p, miR-124-3p, miR-203a-3p and miR-320-3p* in all tested cultures except MG-63. The discrepancies in gene expression pattern between U2 OS, Saos-2 and MG63, as well as lack of characteristic gene profile was described previously and was related to high heterogeneity of the osteosarcoma cell lines [66]. Ths tudy of Pautke et al., also showed that MG-63 is the most heterogeneous population of osteosarcoma cell lines exhibiting both mature and immature osteoblastic features [66], which may explain the obtained gene expression profiles (both mRNA and miRNA) for MG-63 cells.

To the best of our knowledge, the present study, for the first time, highlights that 10 wt % 3 mol % Eu^3+^: nHAp/PLLA scaffolds may act in a selective manner, inducing apoptosis of osteosarcoma cell lines, simultaneously not altering the viability non-transformed cells; here: multipotent stromal cells were isolated from adipose tissue. The scaffolds obtained may find application in theranostics due to luminescent properties of europium, especially in terms of bone healing after cancer resection. The combination of PLLA with nHAp and europium warrants functional performance of the scaffold, thus its potential translation into clinical practice. However, in order to determine the eventual effectiveness of this strategy as an anticancer regimen-modulating bone healing after osteosarcoma resection, further extensive pre-clinical and clinical studies are required.

## Figures and Tables

**Figure 1 materials-12-03779-f001:**
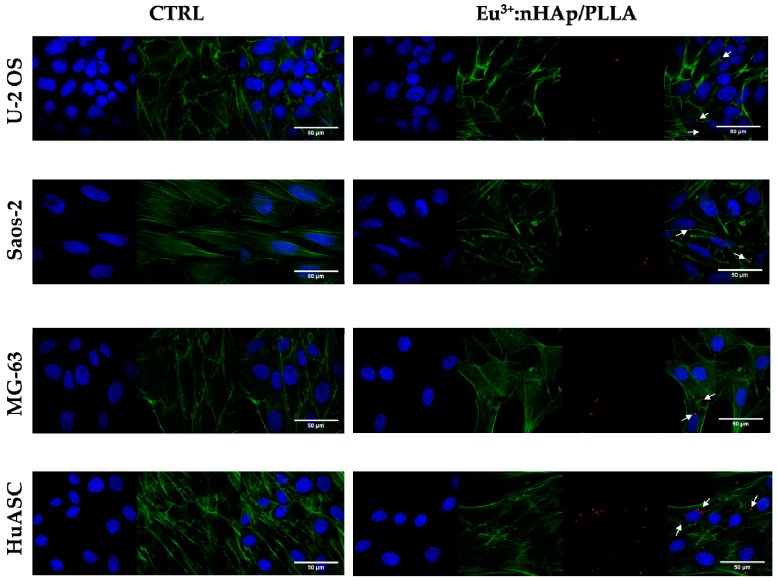
The comparison of cells morphology in control conditions (i.e., on polystyrene/CTRL) and on biomaterial (10 wt % 3 mol % Eu^3+^: nanohydroxyapatite (nHAp)/poly(L-lactic acid) PLLA. The morphology of cells was visualized using confocal microscope. Cells were stained with DAPI (blue, nuclei) and phalloidin atto-488 (green, actin cytoskeleton). Additionally, in research groups the Eu^3+^ ions were visualized (red dots – marked by white cursors). Magnification: 630×, scale bar: 50 µm indicated on merged figure.

**Figure 2 materials-12-03779-f002:**
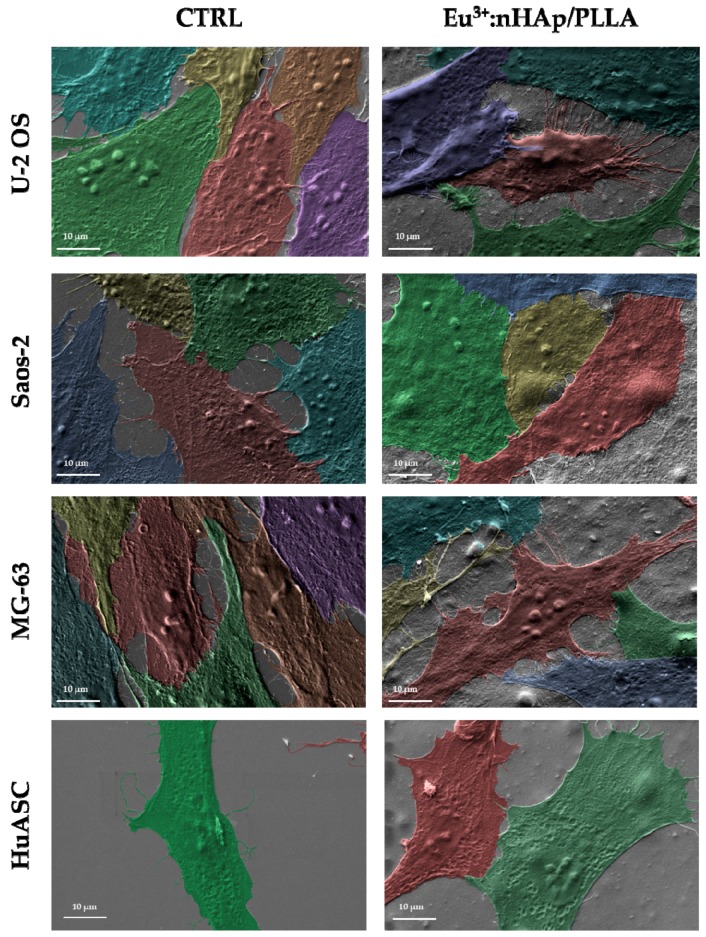
The adhesion and intercellular interactions of cells cultured on polystyrene (CTRL) and biomaterial (10 wt % 3 mol % Eu^3+^: nHAp/PLLA. The cells were visualized using electron microscope (SEM). Magnification: 4000×, scale bar: 10 µm.

**Figure 3 materials-12-03779-f003:**
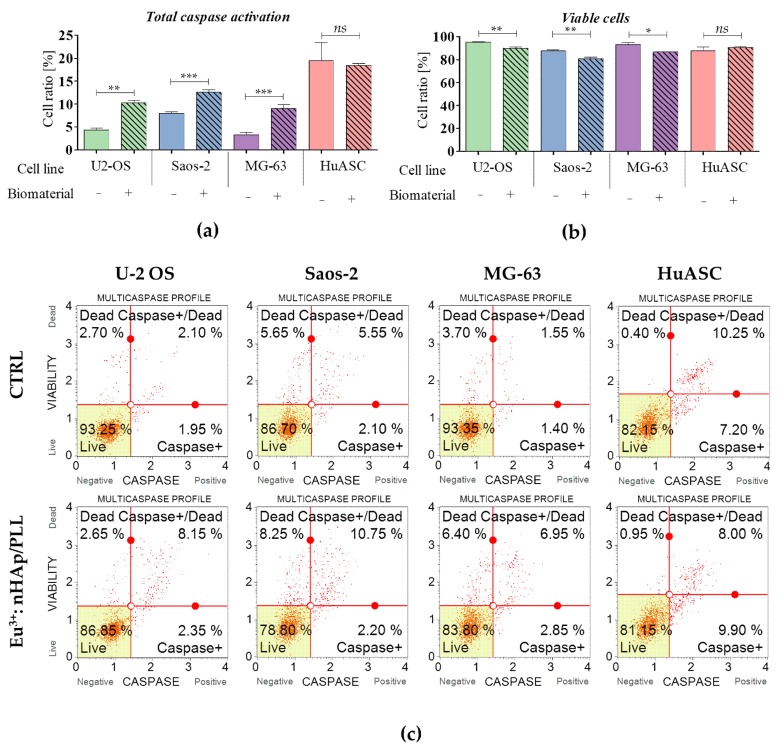
Caspase activity measured in cultures propagated on a polystyrene (CTRL) and on the scaffold (10 wt % 3 mol % Eu^3+^: nHAp/PLLA. (**a**) The comparison analysis of caspase positive cells. (**b**) Comparison analysis of cells’ viability. (**c**) The representative graphs obtained during cytometric-based analysis indicate on cells’ distribution based on caspase activation. Cells were separated into four populations: live (*Live*—bottom left corner), initial activity of caspases (*Caspase +* - down right corner), advanced activity of caspases (*Caspase +* / *Dead*—upper left corner) and dead (*Dead* – upper right corner). The statistically significant differences were marked with an asterisk (*** *p* < 0.001; ** *p* < 0.01, * *p* < 0.05). Non-significant results of comparison are marked as *ns*.

**Figure 4 materials-12-03779-f004:**
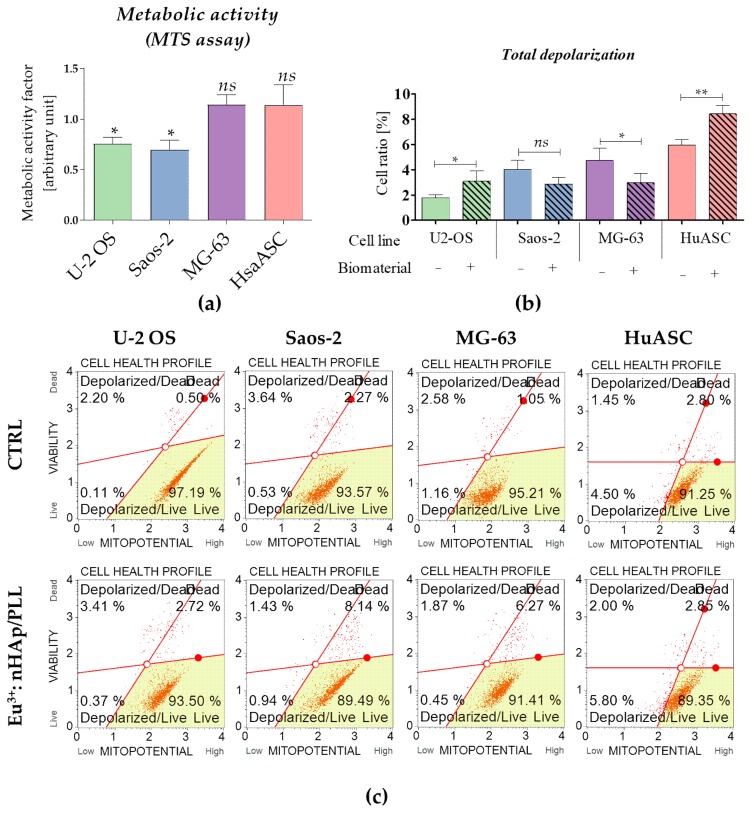
Analysis of cells metabolism in tested culture conditions. (**a**) Metabolic activity was determined using MTS assay. The arbitrary unit reflecting metabolic activity factor was determined for experimental cultures, in reference to the control cultures (considered as 1 = 100% of metabolic activity). (**b**) Mitochondria membrane potential was determined using cytometric-based method. Statistical analysis was performed for the cells with depolarized mitochondrial membrane – total depolarized (live and dead) (**c**) The representative images show distribution of cells based on mitopotential. Cells were separated into four populations: live (*Live*—bottom right corner), with depolarized mitochondrial membrane and live (*Depolarized*/*Live*—bottom left corner), with depolarized mitochondrial membrane and dead (*Depolarized* / *Dead* – upper left corner) and dead (*Dead*—upper right corner). The statistically significant significant differences were marked with an asterisk (*** *p* < 0.001; ** *p* < 0.01, * *p* < 0.05). Non-significant results of comparison are marked as *ns*.

**Figure 5 materials-12-03779-f005:**
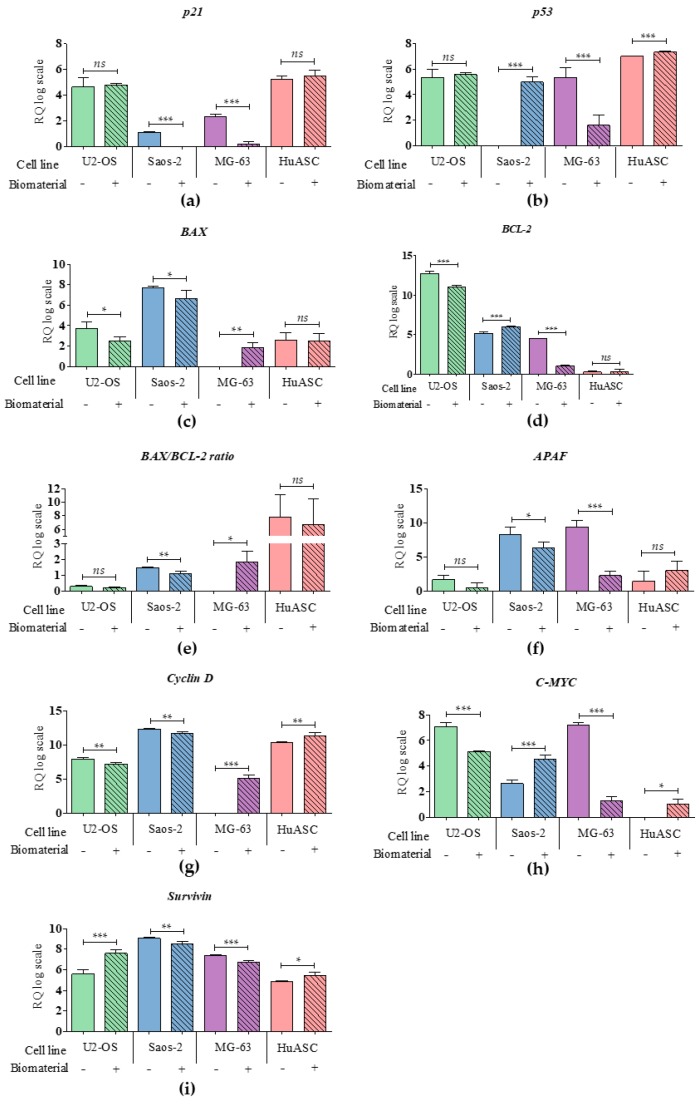
The analysis of mRNA level of genes associated with apoptosis and cell cycle. The examined targets were: (**a**) p21, (**b**) p53, (**c**) BAX, (**d**) BCL-2 with (**e**) BAX/BCL-2 ratio, (**f**) APAF, (**g**) Cyclin D, (**h**) C-MYC, (**i**) Survivin. An asterisk marks a statistically significant differences determined between tested culture circumstances (* *p* < 0.05, ** *p* < 0.01, *** *p* < 0.001).The expression of genes were measured using the RT-qPCR technique, the relative quantification (RQ) was performed using 2^-DDCt^ method and scaling to the sample with highest expression of target gene. The results are expressed as mean ± SD.

**Figure 6 materials-12-03779-f006:**
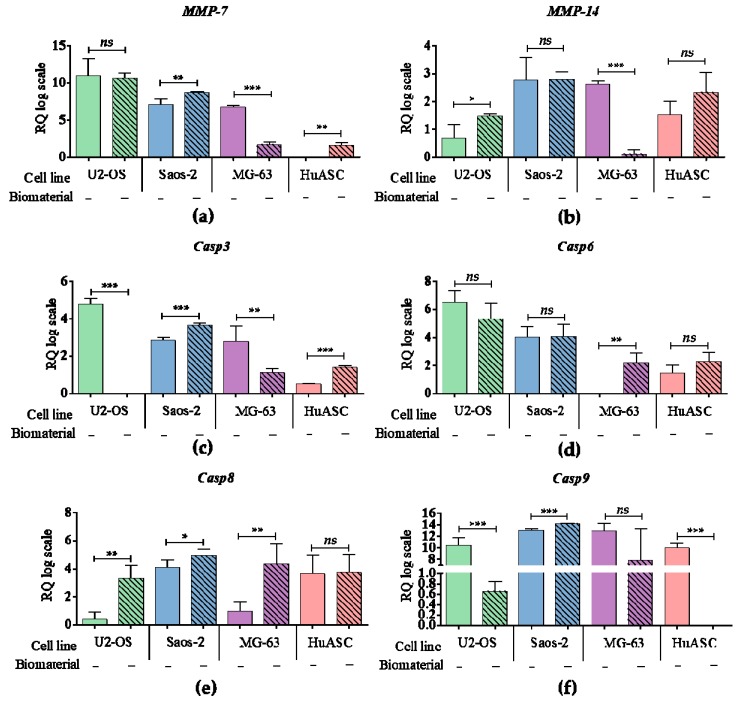
The analysis of mRNA level of genes associated with apoptosis and cell cycle. The examined targets were: (**a**) *MMP-7,* (**b**) *MMP-14* and caspases: (**c**) *caspase 3,* (**d**) *caspase 6,* (**e**) *caspase 8* and (**f**) *caspase 9*. An asterisk marks a statistically significant differences determined between tested culture circumstances (* *p* < 0.05, ** *p* < 0.01, *** *p* < 0.001). The expression of genes were measured using RT-qPCR technique, the relative quantification (RQ) was performed using 2^-DDCt^ method and scaling to the sample with highest expression of target gene. The results are expressed as mean ± SD.

**Figure 7 materials-12-03779-f007:**
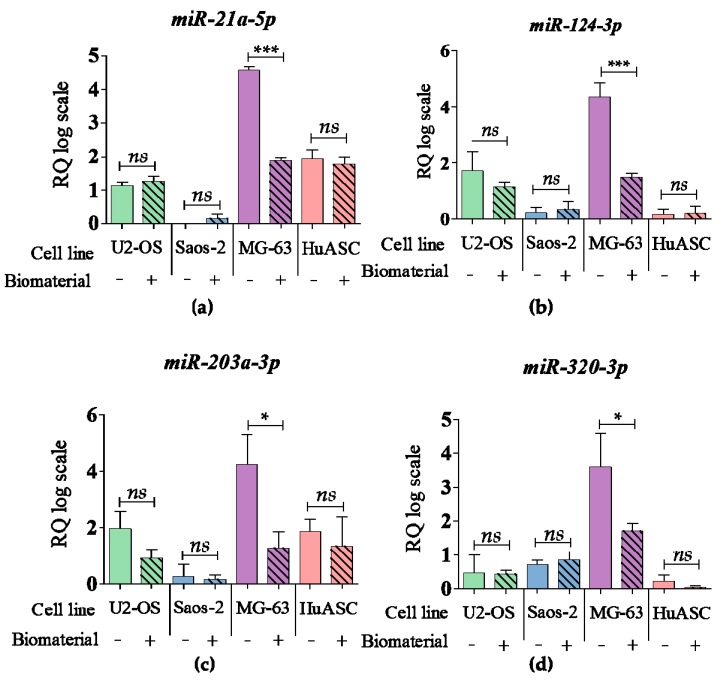
The analysis of miRNA level. The examined targets were: (**a**) *miR-21a-5p,* (**b**) *miR-124-3p,* (**c**) *miR-203a-3p,* (**d**) *miR-320-3p.* An asterisk marks a statistically significant differences determined between tested culture circumstances (* *p* < 0.05, ** *p* < 0.01, *** *p* < 0.001). The expression of miRNA were measured using the RT-qPCR technique, the relative quantification (RQ) was performed using 2^-DDCt^ method and scaling to the sample with highest expression of target gene. The results are expressed as mean ± SD.

**Table 1 materials-12-03779-t001:** List of primers used in quantitative reverse transcriptase real-time polymerase chain reaction (RT-qPCR) for mRNA.

Gene	Primer Sequence 5’-3’	Loci	Amplicon Lenght [bp]	Accesion No.
*P21*	**F:**GGCAGACCAGCATGACAGATTTC	705-727	72	NM_001291549.1
	**R:**CGGATTAGGGCTTCCTCTTGG	776-756		
*P53*	**F:**AGATAGCGATGGTCTGGC	868-885	381	NM_001126118.1
	**R:**TTGGGCAGTGCTCGCTTAGT	1229-1248		
*BCL-2*	**F:**ATCGCCCTGTGGATGACTGAG	1010-1030	129	NM_000633.2
	**R:**CAGCCAGGAGAAATCAAACAGAGG	1138-1115		
*BAX*	**F:**ACCAAGAAGCTGAGCGAGTGTC	235-256	414	NM_001291428.1
	**R:**ACAAAGATGGTCACGGTCTGCC	648-627		
*CYCLIN D*	**F:**GATGCCAACCTCCTCAACGA	264-283	211	NM_053056.2
	**R:**GGAAGCGGTCCAGGTAGTTC	474-455		
*C-MYC*	**F:**CTTCTCTCCGTCCTCGGATTCT	1847-1868	204	NM_001354870.1
	**R:**GAAGGTGATCCAGACTCTGACCTT	2050-2027		
*SURV*	**F:** ACCGCATCTCTACATTCAAG	114-143	113	NM_001168.3
	**R:** CAAGTCTGGCTCGTTCTC	226-209		
*MMP-7*	**F:**TGTATGGGGAACTGCTGACA	488-507	151	NM_002423.5
	**R:**GCGTTCATCCTCATCGAAGT	638-619		
*MMP-14*	**F:** TCGGCCCAAAGCAGCAGCTTC	312-332	180	NM_004995.4
	**R:** CTTCATGGTGTCTGCATCAGC	491-471		
*APAF*	**F:**CTTCTTCCAGTGTAAGGACAGT	861-882	243	NM_013229.2
	**R:**CTGAAACCCAATGCACTCCC	1103-1084		
*CASP3*	**F:**AATACCAGTGGAGGCCGACT	650-669	128	NM_001354779.1
	**R:**TGTCGGCATACTGTTTCAGC	777-758		
*CASP6*	**F:**TCATGAGAGGTTCTTTTGGCAC	231-252	197	NM_001226.3
	**R:**CACACACAAAGCAATCGGCA	427-408		
*CASP8*	**F:**TGCTGAGCACGTGGAGTTAG	282-301	178	NM_001080125.1
	**R:**CAGGCTCAGGAACTTGAGGG	459-440		
*CASP9*	**F:**CTGCGTGGTGGTCATTCTCT	763-782	130	NM_032996.3
	**R:**GCAGCTGGTCCCATTGAAGA	892-873		
*GAPDH*	**F:** GTCAGTGGTGGACCTGACCT	894-913	256	NM_001289746.1
	**R:** CACCACCCTGTTGCTGTAGC	1149-1130		

**Table 2 materials-12-03779-t002:** List of primers used in RT-qPCR for miRNA.

Gene	Primer Sequence 5’-3’	Loci	Accesion No.
*miR-21a-5p*	UAGCUUAUCAGACUGAUGUUGA	18-39	MIMAT0000530
*miR-124-3p*	UAAGGCACGCGGUGAAUGCC	44-63	MIMAT0000134
*miR-203a-3p*	GUGAAAUGUUUAGGACCACUAG	65-86	MI0000283
*miR-320-3p*	AAAAGCUGGGUUGAGAGGGCGA	48-69	MI0000704

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
