# Peer review of "The Potential Selective Cytotoxicity of Poly (L- Lactic Acid)-Based Scaffolds Functionalized with Nanohydroxyapatite and Europium (III) Ions toward Osteosarcoma Cells"

_materials, 2019, doi:10.3390/ma12223779_

Round 1
Reviewer 1 Report
1. Please cite:
X. Wang, M. Lin,Y. Kang, Engineering Porous β-Tricalcium Phosphate (β-TCP) Scaffolds with Multiple Channels to Promote Cell Migration, Proliferation, and Angiogenesis, ACS applied materials & interfaces, 11 (2019), pp. 9223-9232.
M. Lin, N. Firoozi, C.-T. Tsai, M. B. Wallace,Y. Kang, 3D-printed flexible polymer stents for potential applications in inoperable esophageal malignancies, Acta biomaterialia, 83 (2019), pp. 119-129.
2. Fig. 3c and 4a are not clear. The figure's format is not consistent.
Reviewer 2 Report
It is a very good job, well written and described. Although there are some things that have to be taken noticed and written.
Firstly, it is good the novelty of this paper to be described in the introduction. Second, include more relevant and resent references in the paper Third, the discussion of the experimental results should be more detailed, and agreement and disagreement of this work with the work of other authors cited in the references should be also discussed. Finally, the conclusion should be improved.
Reviewer 3 Report
Dear Authors,
The following points in the manuscript could be revised:
Abstract: A contextualization and background of the described work could be add.
Line 45: Clarify “The latest data”.
Line 52: The bibliographic source of the sentence “Nowadays, traditional methods of osteosarcoma treatment are primarily relying on tumor and metastases removal” is the references [9,10]? The authors could mention the methods used for the removal including surgical margins.
Line 57: Describe the effects on the “cells of mesenchymal origin of the most common used drugs.
Line 84: Justify why the screening assays were not shown.
Line 141: Mention the manufacturers of the 10% of fetal bovine serum (FBS) and the 1% of antibiotic solution.
Line 142: Have the “1% of antibiotic solution” included antifungal active principle?
Line 158: Describe the method of cell seeding (eg cell number per scaffold) and culture of the studied scaffolds before fixed for further microscopical analysis.
Line 174: Standardize the use of the symbol ® in the manufacturers’ names.
Line 236: Clarify the sentence: “The alteration of actin cytoskeleton effects on weakened intercellular interactions.“
Line 379: Correct “Bastos”.
In the Discussion section, the authors could give examples of scaffolds studied for cancer regenerative medicine and also describe the potential clinical application of these biomaterials in a practical point of view.
There are a few orthographic and grammatical errors to be reviewed.
Round 2
Reviewer 1 Report
This manuscript has been improved after the reviewing step. I recommend the publication of the manuscript.
Reviewer 3 Report
Dear Authors, I believe that this manuscript materials-629994, entitled "The potential selective cytotoxicity of poly(l-lactid)acid)-based scaffolds functionalized with nanohydroxyapatite and europium (III) ions, toward osteosarcoma cells.” could be recommended as acceptable for publication in the Materials.